# Subtomogram averaging of COPII assemblies reveals how coat organization dictates membrane shape

Joshua Hutchings[1], Viktoriya Stancheva[2], Elizabeth A. Miller [2] & Giulia Zanetti [1]

Eukaryotic cells employ membrane-bound carriers to transport cargo between compartments in a process essential to cell functionality. Carriers are generated by coat complexes that couple cargo capture to membrane deformation. The COPII coat mediates export from the endoplasmic reticulum by assembling in inner and outer layers, yielding carriers of variable shape and size that allow secretion of thousands of diverse cargo. Despite detailed understanding of COPII subunits, the molecular mechanisms of coat assembly and membrane deformation are unclear. Here we present a 4.9 Å cryo-tomography subtomogram averaging structure of in vitro-reconstituted membrane-bound inner coat. We show that the outer coat (Sec13–Sec31) bridges inner coat subunits (Sar1–Sec23–Sec24), promoting their assembly into a tight lattice. We directly visualize the membrane-embedded Sar1 amphipathic helix, revealing that lattice formation induces parallel helix insertions, yielding tubular curvature. We propose that regulators like the procollagen receptor TANGO1 modulate this mechanism to determine vesicle shape and size.

[1] Institute of Structural and Molecular Biology, Birkbeck College, Malet St., London WC1E 7HX, UK. [2] MRC Laboratory of Molecular Biology, Francis Crick Ave., Cambridge CB2 0QH, UK. Correspondence and requests for materials should be addressed to G.Z. (email: g.zanetti@mail.cryst.bbk.ac.uk)

Thousands of newly synthesized proteins exit the endoplasmic reticulum (ER) via COPII-coated carriers. The COPII coat assembles into inner and outer layers to promote membrane deformation and budding, concomitant with cargo capture. The COPII inner coat comprises the small G-protein Sar1 and Sec23–Sec24 heterodimers, while the outer coat is formed by Sec13–Sec31 heterotetramers[1,2]. GTP binding on Sar1, promoted on the ER membrane by the GDP/GTP exchange factor (GEF) Sec12, exposes an N-terminal amphipathic helix that inserts into the membrane, inducing curvature and initiating budding[3]. Membrane-bound Sar1 in turn recruits Sec23–Sec24. Sec24 binds cargo, while Sec23 acts as a GTPase-activating protein (GAP) for Sar1, and also recruits the outer coat proteins Sec13–Sec31 (ref. [2]). Rod-shaped Sec13–Sec31 tetramers assemble into cage-like structures that can adopt different geometries, but always form through interaction of four N-terminal Sec31 β-propeller domains[4–6]. Sec31 contacts Sec23 and Sar1 via a portion of its disordered C-terminus, accelerating Sec23 GAP activity and promoting coat disassembly[7,8]. In addition to the GAP-accelerating peptide, the Sec31 C-terminus contains multiple triple-proline motifs. These motifs bind the gelsolin-like domain of Sec23 (ref. [9]) and are shared among other Sec23-binding proteins including the scaffolding protein Sec16 (refs. [10,11]) and the procollagen receptors TANGO1 and cTAGE5 (ref. [9]).

In vitro, the COPII coat can be stably retained on membranes by incubation with non-hydrolyzable GTP analogs, yielding coated carriers with varied morphologies, including spheres and tubules[8]. On tubules, Sar1–Sec23–Sec24 forms extended ordered arrays, indicative of a structural role in carrier morphogenesis[4,9,12]. Straight or constricted tubules have also been seen in cells, indicating that COPII is able to generate carriers with tubular morphology in vivo[13,14]. This is likely important for traffic of large cargo like procollagens, which can reach 300 nm in length[2]. Accessory factors such as TANGO1 have been proposed to modulate the mechanics of vesicle formation to generate carriers appropriate for procollagen transport[9].

How membrane carriers of different shapes and sizes can be generated through regulation of coat assembly has remained an open question. To gain a mechanistic understanding of how coat assembly drives membrane deformation, we used cryo-tomography and subtomogram averaging to obtain detailed structural information on COPII assembled on a lipid bilayer. We obtained the structure of the assembled inner COPII coat to 4.9 Å resolution, representing a significant advance on the previously solved structure of the assembled COPII coat[4], and the highest resolution subtomogram average of a membrane-bound coat to date. Our structure reveals in molecular detail how outer coat proteins promote inner coat assembly through their triple-proline motifs, and how parallel orientation of inner coat subunits and insertion of the Sar1 amphipathic helix promotes tubular curvature of budded membranes. We suggest a mechanism for regulation of membrane curvature and shape, based on the organization of the inner coat by the outer coat and accessory factors.

## Results

**Binding of Sec13–Sec31 is necessary for inner coat assembly**. We reconstituted COPII budding from giant unilamellar vesicles (GUVs) using purified yeast proteins in the presence of a non-hydrolyzable GTP analog, GMP-PNP. Reconstitution reactions revealed tubules with ordered arrays of inner and outer coat, as seen previously[4] (Fig. 1a and Supplementary Figure 1a). Formation of Sar1–Sec23–Sec24 arrays depends on the outer coat: a reaction lacking Sec13–Sec31 yielded coated GUVs with some local deformation but no discernable order to the inner coat

(Supplementary Figure 1b) despite robust Sar1–Sec23–Sec24 recruitment (Supplementary Figure 1c). We also reconstituted budding using Sec31 tagged with an N-terminal 6xhis peptide that was not cleaved (Sec31-Nhis). Sec31-Nhis was recruited well to liposomes indicating that it retains fundamental binding functions (Supplementary Figure 1c). In reconstituted budding reactions we observed few coated vesicles and many tubes with a range of diameters (Supplementary Figure 2). On these tubes the outer coat was disordered: in contrast to the ordered lozenges seen with wild-type Sec31, we could clearly visualize rods that appeared to "float", tethered to the budded tubule without forming regular arrays (Fig. 1b and Supplementary Figure 1d). We hypothesize that the peptide tag immediately adjacent to the N-terminus perturbs stability of the Sec31 β-propellers or their interaction interface, and thus disfavors cage formation. In these conditions, the inner coat clearly formed ordered lattices (Fig. 1b and Supplementary Figure 1d), indicating that inner coat assembly requires binding of Sec31 but is not dependent on its persistent polymerization, and suggesting that the inner coat has a major structural role in COPII vesicle formation. Sec31-Nhis supported in vitro budding from microsomes in the presence of GMP-PNP, but in the presence of GTP failed to generate vesicles (Fig. 1d). Thus, under conditions that permit dynamic coat turnover, a stable outer coat assembly is required for productive budding (Fig. 1d). Interestingly, Sec31-Nhis supported viability in yeast (Fig. 1c), suggesting that additional factors may stabilize the coat in cells.

**Subtomogram average of inner COPII at 4.9 Å resolution**. Cryo-tomography and subtomogram averaging on tubules formed in the presence of Sec31-Nhis yielded a structure of membrane-associated Sar1–Sec23–Sec24 trimers at an average resolution of 4.9 Å (Fig. 2, Supplementary Figure 3 and Supplementary Movie 1). Placing the average structure onto its original positions and orientations in tomograms confirmed the formation of arrays of inner coat, which either completely coated uniform tubes or formed less extended patches compatible with changes in tube diameter and curvature (Fig. 2a, b and Supplementary Figure 4). The unit particle we used for subtomogram averaging contained three complete Sar1–Sec23–Sec24 heterotrimers, which could be fitted with the X-ray structures of the Sar1–Sec23 and Sec23–Sec24 complexes determined previously[15] (Fig. 2c–f). Since the local resolution in the majority of our cryo-EM map ranged from 4.5 to 7.5 Å (Supplementary Figure 3), we decided to avoid refinement of the atomic model. As seen before[4], we confirmed that lateral interactions between inner coat trimers occur mainly via Sec23–Sec23 interfaces (Supplementary Figure 5a), with additional contributions from Sar1–Sec23 interfaces (Supplementary Figure 5b). Sec24 molecules also contact each other, albeit less extensively.

**Interaction between Sar1 and the membrane**. The membrane bilayer was visible in our cryo-EM map (Fig. 3a, Supplementary Figure 6a–c), and we could clearly see strong density protruding from the Sar1 N-terminus and inserting into the membrane outer leaflet (Fig. 3b and Supplementary Movie 1). We can confidently attribute the bulk of that density to the Sar1 N-terminal amphipathic helix. Its clearly defined elongated shape indicates that Sar1 amphipathic helices all adopt roughly the same orientation with respect to the protein core, running along the long axis of the tube (Fig. 3c). Inspection of the EM density shows that the Sar1 N-terminal region inserts ~10–12 Å into the membrane outer leaflet, and then takes a sharp, roughly 90°, bend (Fig. 3b, and Supplementary Figure 6a–c). Rosetta algorithms[16] predict a kink at positions G18-L19 of the helix, which is fully consistent with the

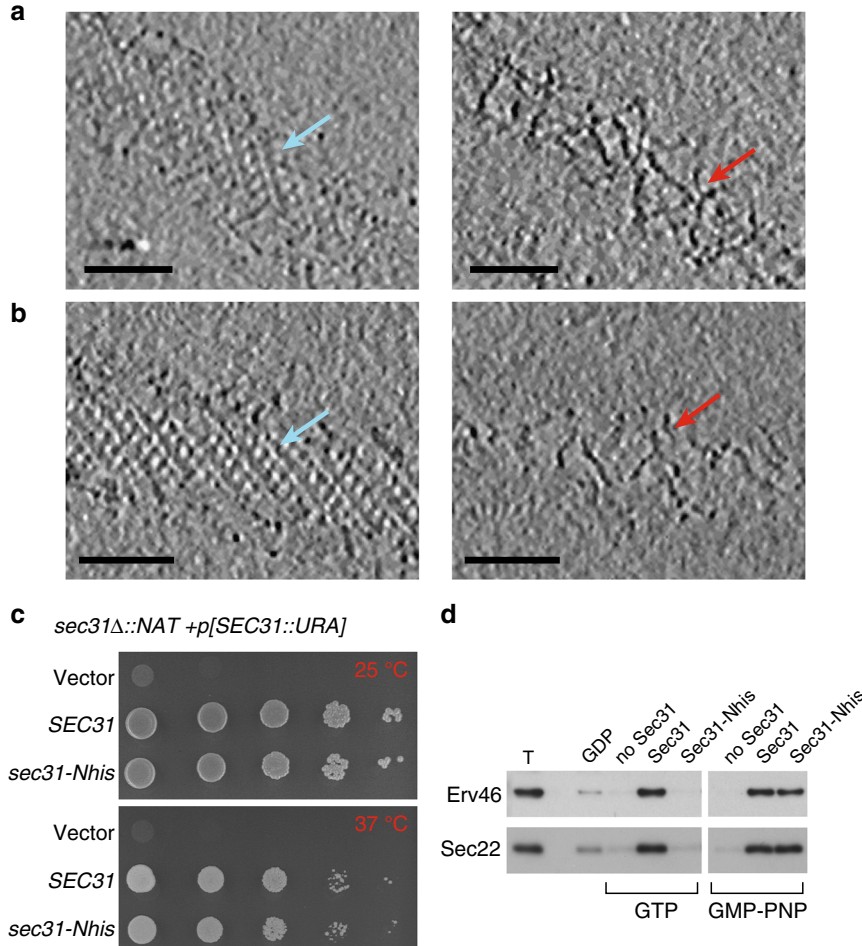

**Fig. 1** Inner coat assembly depends on outer coat binding. **a** XY slices through a tomogram of budding reactions reconstituted with wild-type Sec13–Sec31 (Sec31 C-terminally his-tagged), showing regular inner coat (blue arrow) and outer coat (red arrow). Scale bars: 50 nm. **b** XY slices through a tomogram of budding reactions reconstituted with Sec13/31-Nhis, showing regular inner coat (blue arrow) but not outer coat (red arrow). **c** A yeast strain deleted for *SEC31*-bearing plasmids as indicated was tested for viability at 25 and 37 °C, revealing robust growth at both temperatures. **d** In vitro COPII budding reactions from microsomal membranes were performed using GDP/GTP/GMP-PNP and Sec31/Sec31-Nhis as indicated. T corresponds to 10% of the total donor membranes; remaining lanes correspond to recovered vesicles. Packaging of two cargo proteins, Erv46 and Sec22, was measured by immunoblotting

cryo-EM density (Supplementary Figure 6b). This orientation is compatible with models proposing that amphipathic helices embed shallowly into lipid bilayers, creating a wedge to induce and/or sense membrane curvature[17]. Membrane curvature likely derives from Sar1 amphipathic helix insertion, as well as interaction with the membrane-proximal surface of Sec23–Sec24. Sec24 in particular appears to extensively contact the membrane through a largely basic surface that encompasses its Zn finger domain (Supplementary Figure 6d–e). Formation of ordered arrays of inner coat thus leads to many Sar1 helices inserting parallel to each other (Fig. 3c) and to the inner surface of Sec23–Sec24 to adopt a cylindrical profile, reinforcing local membrane curvature and favoring tubes. This might in turn be sensed by additional Sar1 and Sec23–Sec24 subunits, which are efficiently recruited to the site of budding to further promote tubulation.

**Mechanism for organization of the inner coat by Sec31.** To address how Sec13–Sec31 promotes assembly of the inner coat without itself being ordered, we sought to identify regions of density associated with the inner coat but that could not be explained by its atomic model. Such density might be attributable

to outer coat binding. We calculated a difference map between the fitted atomic model and the cryo-EM map, and applied a Gaussian filter to identify the largest regions of continuous density (Fig. 4 and Supplementary Movie 1). Four distinct regions emerged: the membrane with the embedded Sar1 helix; a region of density protruding from the gelsolin domain of Sec24 that was poorly resolved in the crystal structure[15] (Fig. 4a, b, dashed box); and two regions distal to the membrane that could not be attributed to inner coat proteins (Fig. 4a, b, pink and orange boxes). The first unassigned density is located at the interface between Sar1 and Sec23 (Fig. 4c). This likely corresponds to residues within the catalytic peptide of Sec31, previously defined by X-ray crystallography[8]. Within this region, Sec31-W922, -N923, and -L925, locate to the interface between Sec23 and Sar1 and are important for GAP acceleration by Sec31 (refs. [8]). Superposing the X-ray structure of the Sar1–Sec23 complex co-crystallized with the active fragment of Sec31 confirmed that the extra density in our map is in proximity to the catalytic residues, strongly suggesting that these residues are tightly bound to Sec23-Sar1, poised to enhance GTP hydrolysis (Fig. 4c). The cryo-EM density does not completely overlap with the X-ray model of the Sec31 active fragment, perhaps due to minor conformational rearrangements. The second unassigned density is located

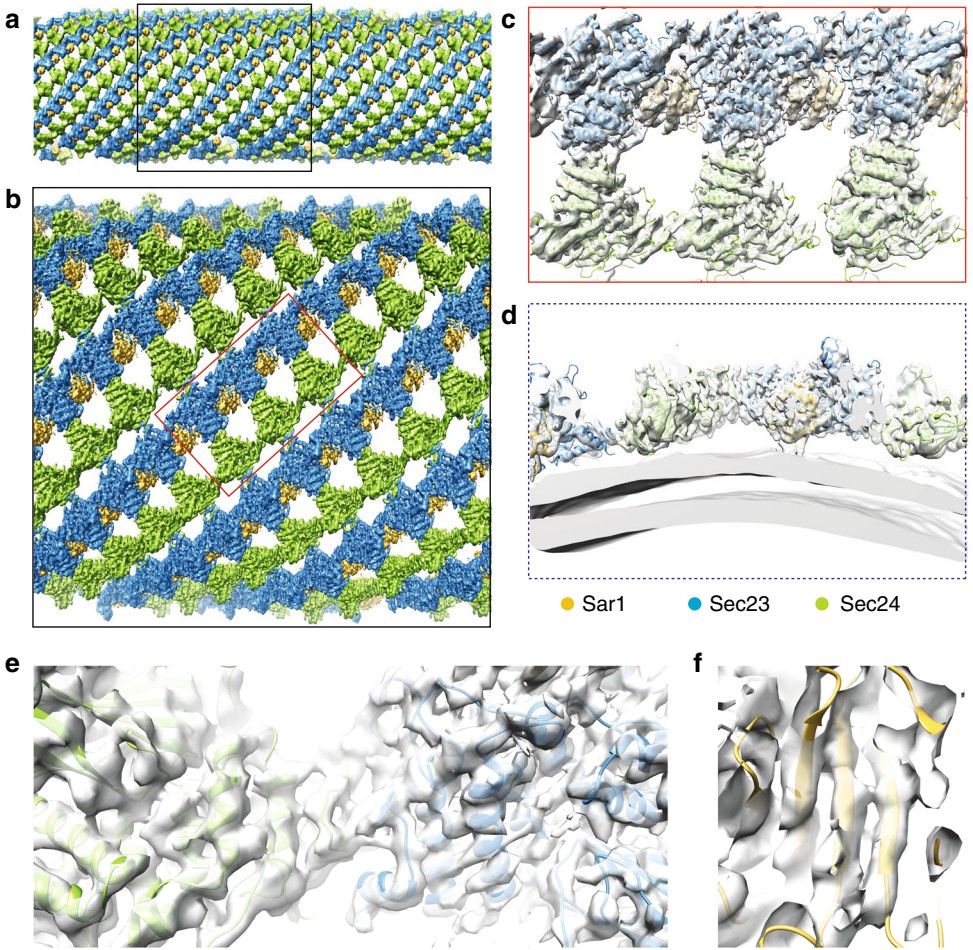

**Fig. 2** 4.9 Å cryo-tomography structure of the assembled inner coat. **a** Subtomogram averages are back-plotted according to their aligned positions and orientations, showing a regular helical arrangement. Sec23 is blue, Sec24 is green, and Sar1 is yellow. **b** Zoomed region of **a**. **c**, **d** Surface representation of the locally filtered cryo-EM map (transparent white), with the atomic models (PDB 1M2O and 1M2V) fitted and colored according to the scheme in panel **a**. Top and side views, respectively. The membrane in panel **d** was segmented from a Gaussian-filtered version of the structure. **e**, **f** Details of the cryo-EM map that show the quality of the fit

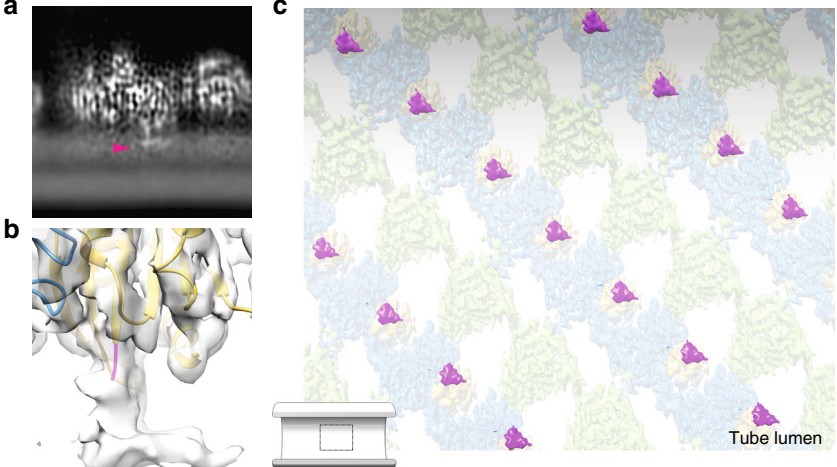

**Fig. 3** Sar1 amphipathic helix membrane insertion. **a** XZ slice through the cryo-EM density map, showing the lipid bilayer, and a strong density protruding from the protein core and inserting into the outer leaflet (magenta arrowhead). **b** Surface representation of the membrane inserted region shows continuity with the N-terminus of Sar1 X-ray structure, which lacks the Sar1 amphipathic helix. The N-terminal residue of the X-ray structure is depicted in magenta. **c** The back-plotted density is shown from the inside of a tube, highlighting the Sar1 amphipathic helix in magenta

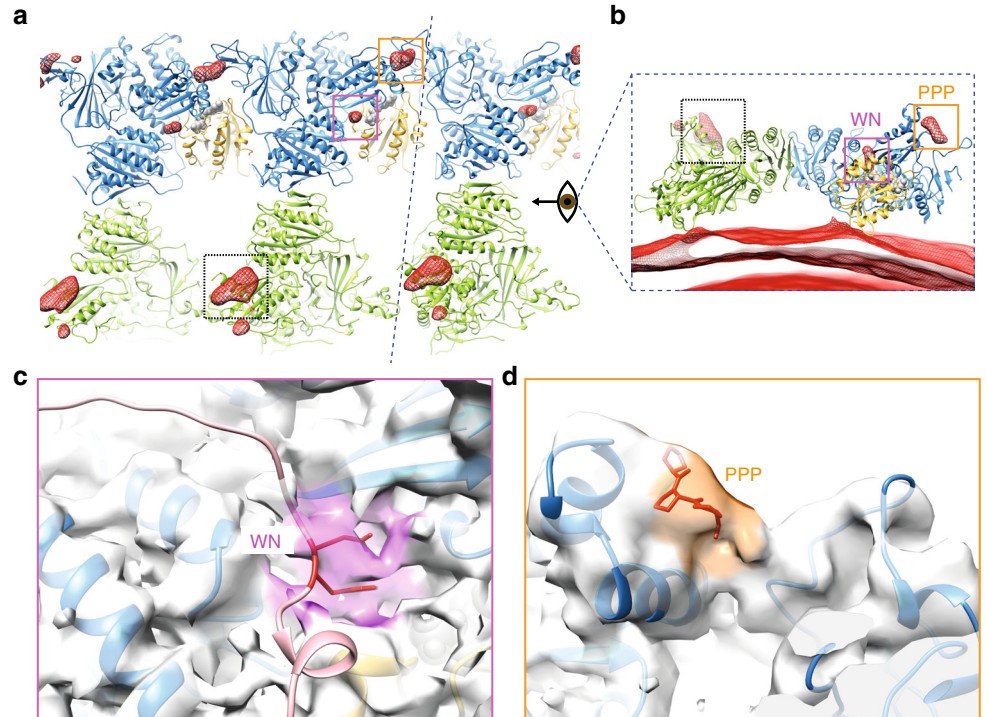

**Fig. 4** Binding of the outer coat. **a** Fitted inner coat atomic model superimposed to its difference map with the cryo-EM density (Gaussian filtered). Dashed box: a region on Sec24 that was not resolved in the X-ray structure. Pink box: unassigned extra density corresponding to the Sec31 catalytic residues (WN) binding site. Orange box: unassigned extra density corresponding to the Sec31 tri-proline binding site. **b** Side view of **a**, where the membrane is also visible in the difference density map. **c** Detailed view of the region in the pink box, where the atomic model has been overlaid with the X-ray structure of the Sec31 active fragment (light pink, PDB 2QTV), and the density unoccupied by Sar1 and Sec23 is shaded in pink. Residues WN (922–923) are highlighted in red. **d** Detailed view of the region in the orange box, where the atomic model has been overlaid with the X-ray structure of the TANGO1 PPP motif (red, PDB 5KYW), and the density unoccupied by Sec23 is shaded in orange

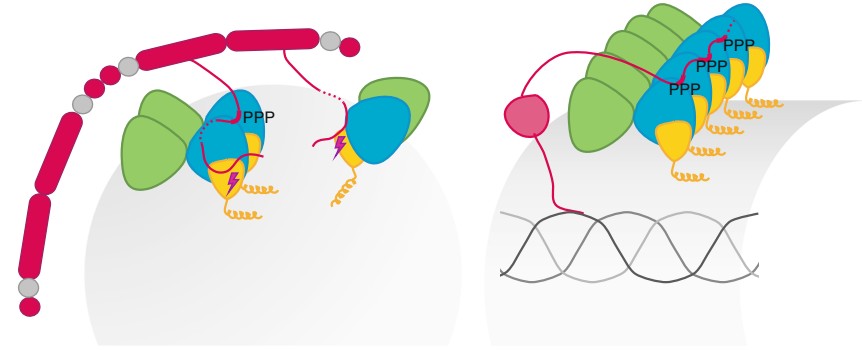

**Fig. 5** Model for regulation of COPII carrier size. A model that summarizes the findings in this paper. On the left: Sec31 (red) binds to Sec23 (blue) through its PPP regions inducing inner coat assembly at the same time it promotes GTP hydrolysis and coat disassembly by binding through its catalytic peptide. This leads to small inner coat patches that insert Sar1 amphipathic helices randomly and curve the membrane in all directions. On the right: TANGO1 (pink) promotes inner coat assembly through PPP binding, without inducing GTP hydrolysis, while at the same time binding to procollagen molecules in the ER lumen. This leads to extensive assembly of the inner coat, consistent Sar1 orientation and parallel insertion of its amphipathic helix, and membrane tubulation, which promotes procollagen export

between neighboring Sec23 molecules at the inner coat lattice interface (Fig. 4d). This density lies adjacent to the gelsolin domain of Sec23, a region recently identified as a site of interaction between human Sec23 and PPP-containing peptides of TANGO1 (ref. [9]). We superimposed the hSec23-TANGO1-PPP co-crystal structures to our fitted atomic model of Sec23, which revealed correspondence between the extra density and the PPP binding site (Fig. 4d). Thus, our structure suggests that Sec31 binds to Sec23 through PPP motifs at the same site as TANGO1, consistent with biochemical experiments that showed

competition between the two proteins for Sec23 binding[9]. Our assembled structure reveals that the PPP motif and adjacent residues form a bridge connecting neighboring Sec23 subunits: the PPP residues are part of a more extended region of density, which appears connected to both Sec23 molecules. The resolution in this area is lower than the rest of Sec23 (Supplementary Figure 3), likely because different PPP-containing regions (yeast Sec31 has seven such motifs) are bound to distinct Sec23 molecules.

## Discussion

Our structure reveals that Sec31 directly contributes to lateral interactions between adjacent Sec23 molecules via binding through its PPP motifs and adjacent sequences. Stable polymerization of Sec13–Sec31 into ordered cage-like structures is not absolutely required for this influence on inner coat oligomerization. However, under conditions of coat turnover, Sec31 oligomerization is important, since Sec31-Nhis cannot support vesicle formation from microsomal membranes in the presence of GTP (Fig. 1d). Sec31 likely binds to Sec23 simultaneously both via its PPP motifs and its catalytic residues, meaning that kinetics of coat assembly (via PPP motifs) and disassembly (via GTP hydrolysis) can be directly modulated by Sec31 to yield a productive budding event. When GTP hydrolysis is permitted, inner coat lateral interactions are transient, yielding Sar1 amphipathic helices that orient randomly and thereby drive membrane curvature in all directions, resulting in a spherical vesicle with no extended inner coat lattice. Under conditions of limited GTP hydrolysis, inner coat assembly would predominate over disassembly, and more extended lattices with parallel orientation of multiple Sar1 helices would lead to a uniform direction of membrane curvature and thus formation of tubules in a reaction independent of outer coat stable assembly. As proposed previously[9,18], TANGO1 might promote inner coat assembly by blocking recruitment of Sec13–Sec31 and thereby inhibiting its GAP stimulation while simultaneously positioning Sar1–Sec23–Sec24 elements favorably for lattice formation (Fig. 5). Interestingly, cTAGE5, a TANGO1 co-receptor that contributes to ER export of procollagen, interacts with Sec12, possibly further shifting the balance of GTP- versus GDP-bound Sar1 toward more stable coat assemblies[19]. Other Sec23-interacting proteins might also use similar mechanisms to promote coat assembly. Sec16 is an essential protein located at sites of COPII budding that acts as a scaffold to promote COPII recruitment[10,11]. Like TANGO1, Sec16 also impedes recruitment of Sec13–Sec31 to the inner coat[20], and contains proline-rich domains that encompass multiple interspersed PPP motifs.

In summary, cryo-electron tomography and subtomogram averaging revealed the molecular mechanisms of regulation of COPII coat assembly, and how these are reflected in determination of membrane shape. Many other membrane trafficking events involve supramolecular assemblies of protein complexes that contain amphipathic helices. Regulation of assembly architecture, as we describe here for COPII, might be a widespread mechanisms to impose a particular curvature onto the underlying membrane.

## Methods

**Cloning**. All yeast proteins were cloned from the *Saccharomyces cerevisiae* S288c strain genome into appropriate expression vectors using In-Fusion (Takara): pETM-11 for Sar1 and pFASTBacHTb for Sec23/24 and Sec13/31. All primers used here are summarized in Supplementary Table 1.

**Expression and purification of Sar1**. An N-terminal His-tagged Sar1 construct was expressed in BL21 *E. coli* cells (Star™ DE3; Invitrogen), and affinity purified by binding to HisTrap (GE Healthcare) column in buffer A (50 mM TRIS buffer with 150 mM NaCl, 0.1% Tween-20 (v/v), 10 mM imidazole, 1 mM DTT, pH 8.0), and

eluting with an imidazole gradient. His-tagged TEV protease cleavage (~1:50 ratio of protease:Sar1 (w/w)) was carried out overnight while dialysing against buffer A. A second HisTrap column was used to retain TEV and cleaved 6His, and Sar1 in the flowthrough was concentrated using a 10 kDa MWCO centrifugal filter to 0.85 mg/ml (~38.6 μM), ready for use in budding reactions.

**Expression and purification of Sec23/24**. One liter of Sf9 insect cells (at $1 \times 10^6$ cells/ml) were infected with 9 ml of baculovirus harboring a bacmid with untagged Sec23 and 3 ml of His-tagged Sec24 virus. The cells were harvested and lysed after 3 days. Sec23–Sec24 heterodimers were affinity purified by binding to HisTrap column in lysis buffer (20 mM HEPES, 250 mM sorbitol, 500 mM potassium acetate, 10 mM imidazole, 10% glycerol, and 1 mM DTT, pH 8.0), and eluting with an imidazole gradient. Fractions containing Sec23–Sec24 were diluted approximately two-fold in low salt buffer (20 mM Tris, 1 mM magnesium acetate, 0.1 mM EGTA, and 1 mM DTT, pH 7.5) prior to loading onto an anion-exchange column (HiTrap Q; GE Healthcare). The complex was eluted using a NaCl gradient, and diluted to a final concentration of ~1.26 mg/ml (~6.6 μM) with low salt buffer plus 10% glycerol prior to flash-freezing and storing at −80 °C.

**Expression and purification of Sec13/31**. One liter of Sf9 insect cells (at $1 \times 10^6$ cells/ml) were infected with 9 ml of untagged Sec13 virus and 3 ml of His-tagged Sec31 virus, either at the N- or C-terminus. HisTrap and anion-exchange was performed as described for Sec23/24. Single frozen aliquots were gel filtrated on a 2.4 ml Superdex200 column (GE Healthcare) equilibrated in the budding reaction buffer (20 mM HEPES, 50 mM KOAc and 1.2 mM MgCl₂, pH 6.8) prior to use in budding reactions. Concentration of the gel filtration peak was measured before each budding reaction.

To produce Sec13/31 with no tag an additional TEV protease cleavage step 1:25 (w/w) was applied following HisTrap elution, followed by overnight dialysis against buffer A. A second HisTrap column was used to retain TEV and cleaved 6His, and Sec13–Sec31 in the flowthrough was applied to an anion-exchange column, frozen and aliquots were gel filtrated as described above.

**Yeast strains and plasmids**. Strains were grown at 30 °C in standard YPAD (yeast extract 11 g/l, peptone 22 g/l, glucose 20 g/l, and adenine sulfate 55 mg/l, supplemented with 100 mg/l nourseothricin (Jena Bioscience) when needed), or synthetic complete media (6.7 g/l yeast nitrogen base, 20 g/l glucose, supplemented with amino acids as needed). If grown on plates 25 g/l agar was included. 5-FOA plates contained 6.7 g/l yeast nitrogen base, 20 g/l glucose, 0.7 g/l synthetic drop-out medium supplement (Sigma-Aldrich), 50 mg/l uracil, 20 g/l agar, and 1 g/l 5-FOA (Sigma). For growth assays SEC31 deletion strain LMY1249 (sec31::NAT pep4::TRP ade2-1 his3-11 leu2-3,112 + [pYCp50::SEC31-URA3]) was transformed with plasmids encoding wild-type SEC31 or sec31-Nhis, and the ability of each form of Sec31 to support growth tested by serial dilution onto media containing 5-FOA to counterselect for the wild-type SEC31::URA plasmid.

**Liposome binding**. Major–Minor Mix lipids[21] were dried to a lipid film in a rotary evaporator and rehydrated to 2 mM HKM Buffer (20 mM HEPES pH 6.8; 160 mM KOAc; 1 mM MgCl₂) at room temperature by occasional vortexing. The resuspended lipids were then extruded 13 times through a 400 nm pore-size polycarbonate filter. Seventy-five microliters binding reactions included lipids and purified COPII components in HKM Buffer to the following final concentrations: 0.27 mM liposomes, 15 μg/ml Sar1, 20 μg/ml Sec23/24, 30 μg/ml Sec13/31, and 0.1 mM Nucleotide. Reactions were incubated for 30 min at 25 °C. Fifty microliters of the reaction volume was mixed with an equal volume of 2.5 M Sucrose-HKM, overlaid with 100 μl 0.75 M Sucrose-HKM and 20 μl HKM. Gradients were spun in a Beckman TLA-100 rotor (100 000 rpm, 20 min, 25 °C). The top 20 μl of the gradient was collected, separated by sodium dodecyl sulfate polyacrylamide gel electrophoresis (SDS-PAGE), and visualized by SYPRO Red.

**Microsome budding assays**. Microsomal membranes (1 mg) were washed with 3 × 1 ml 2.5 M urea-B88 (20 mM HEPES, pH 6.8, 150 mM KOAc, 250 mM sorbitol, 5 mM Mg(OAc)₂) and 3 × 1 ml B88 (15,000 rpm, 2 min, 4 °C). Budding reactions were set up in B88 including 10 μg/ml Sar1, 10 μg/ml Sec23/24, 20 μg/ml Sec13/31, 0.1 mM nucleotide, and ATP Regeneration Mix (final concentrations: 1 mM ATP, 50 μM GDP-mannose, 40 mM creatine phosphate, 200 μg/ml creatine phosphokinase in B88) and incubated (30 min, 25 °C). Twelve microliters of the total reaction was removed and the vesicles separated from the donor membranes by centrifugation (15,000 rpm, 2 min, 4 °C). Vesicles were collected by centrifugation of 200 μl of the supernatant (50,000 rpm, 25 min, 4 °C). The supernatant was discarded and the pelleted vesicles were resuspended in SDS sample buffer and analyzed by SDS-PAGE and immunoblotting with α-Sec22 and α-Erv46 antibodies. Rabbit polyclonal antibodies were raised against a GST-fusion of the soluble cytoplasmic domain of Sec22 and Erv46, respectively. The full immunoblot used in Fig. 1d is supplied in Supplementary Figure 7.

**GUV budding reactions**. GUVs were prepared by electroformation of a major–minor lipid mixture at 10 mg/ml in a 2:1 chloroform/methanol mix[13].

 

Briefly, the lipid mixture was spread over two indium tin oxide (ITO) glass slides that were sandwiched together with a silicon spacer to form a chamber. The chamber was filled with 300 mM sucrose before applying an alternating voltage of 10 Hz and 3 V (rms) for 6–8 h using copper tape. GUVs were harvested by aspirating the sucrose from the chamber and applying to 300 mM glucose at 4 °C overnight.

Size and morphology of GUVs were qualitatively assessed on a Zeiss Axio Scope A.1 fluorescence microscope mounted with a ×63 Zeiss plan-Apochromat lens (1.4 Oil DIC). Budding was reconstituted by adding COPII proteins at defined concentrations (1 μM Sar1, 320 nM Sec23/24, 173 nM Sec13/31), 1 mM GMP-PNP, 2.5 mM EDTA (pH 8.0), and 10% (v/v) GUVs to budding reaction buffer for a total volume of 30 μl. Reactions were gently agitated for mixing and incubated at room temperature for 1–3 h.

**EM sample preparation**. For cryo-EM, 8 μl of 5 nm gold fiducials (BBI solutions) were added to 30 μl budding reaction mixture just prior to vitrification. Four microliters of budding reaction were applied to negatively glow-discharged lacey carbon grids (Agar Scientific, 200 Mesh Copper, S166) and plunge-frozen using an FEI Vitrobot system maintained at 4 °C with 100% humidity. Grids were stored in liquid nitrogen until use in EM.

For negative stain, 4 μl of budding reaction were applied to negatively glow-discharged continuous carbon film, 300 mesh, copper grids (EM Sciences, CF300-Cu), and stained with 2% uranyl acetate following standard procedures. Grids were viewed on the in-house T10 or T12 electron microscopes (Tecnai 100 and 120 kV, respectively) fitted with CCD cameras.

**Cryo-electron tomography data collection**. Budding reactions with Sec31-Nhis yielded a higher number of homogeneous tubes compared to those carried pout with wild-type Sec31, providing an ideal sample for high-resolution structure determination. The tomography dataset of Sec31-Nhis budding reactions were collected at the European Molecular Biology Laboratory (EMBL) cryo-EM facility in Heidelberg on a FEI Titan Krios operated at 300KV. Data were automatically collected using a Gatan Quantum energy filter with a K2 xp detector (Gatan, Pleasanton CA) in EFTEM mode and a 20 eV slit. Pixel size was 1.327 Å, defocus ranged between 1.5 and 3.5 μm. The dose-symmetric tilt scheme[22] implemented in SerialEM[23] was used to collect tilts between −60° and 60°, with 3° increments and a total exposure of ~140$e^-$ Å$^{-2}$. Thirteen superresolution frames were collected per tilt which underwent on-the-fly 4× binning and whole-frame alignment in MotionCor2 (ref. [24]) without dose-compensation. Coma-free alignment was done using FEI AutoCTF. A total of 90 tomograms were collected across a 72 h session, of which 83 were used for subtomogram averaging. The tomogram shown for the wild-type C-his-tagged Sec13/31 construct (Fig. 1a) was collected in-house on a 300 kV Polara microscope fitted with a K2 direct electron detector, energy filter slit at 20 eV, and pixel size of 1.433 Å. A summary of the data collection is given in Supplementary Table 2.

**Cryo-tomography data processing**. Tilt series were initially aligned and reconstructed by weighted back-projection using the IMOD/etomo workflow[25]. Tilt series with residual errors exceeding 1 pixel were discarded. Binned 8X tomograms were initially reconstructed with 50 iterations of SIRT-like filtering for initial inspection, particle picking and alignment. The unbinned, aligned tilt series were dose-filtered using critical exposure values determined in ref. [26], using bespoke MATLAB scripts and a cumulative dose of 2.5$e^-$/Å$^2$ per tilt.

CTF was estimated using CTFFIND4 (ref. [27]), and corrected during back-projection using the NovaCTF pipeline[28]. We found that defocus estimation becomes erratic at high angle tilts, so we used cropped central rectangular regions of the tilt series for the purposes of estimation.

**Particle picking for subtomogram averaging**. The workflow described uses a combination of Dynamo[29] and bespoke Matlab scripts. Initial particle picking was done as described previously[4]. Briefly, ~400,000 binned 8X SIRT-filtered subtomograms were extracted from oversampled coordinates on tube surfaces in 32 voxel boxes. Initial Euler angles were assigned normal to the membrane as determined by the manually traced tube axes. For convenience, the dataset was manually divided in ten batches. The average from the first batch was smeared in the direction of the tube major axis to create a featureless membrane and coat density, used as a starting reference for nine rounds of coarse alignments. The result was used as a reference for initial alignment of the other nine batches upon applying a ~34 Å low-pass filter. Oversampled particles converging onto the same coordinate were removed using Dynamo's separation in tomogram parameter. Per-tube averages were calculated to determine directionality of each tube, and in-plane rotation angles of particles were adjusted accordingly. We used cross-correlation (CC) to eliminate bad particles from the dataset. To compensate for missing wedge-dependent variations in CC, we adjusted the CC values used for thresholding: for each tube, we empirically fit a polynomial to the plot describing CC against the Euler angle phi, and used the coefficients to weight CC values. This way, subtomograms that clustered around gold beads or that converged in irregular patterns away from the tube axis became clear outliers and were easily thresholded

away. A combined dataset of ~100,000 particles was obtained, which was further divided between two half datasets (A and B) for independent processing.

**Alignments and subtomogram averaging**. CTF corrected subtomograms were extracted and aligned to low-pass filtered references using Dynamo. Iterations were carried out starting from binned 8X data, using a low-pass filter of 34 Å, angular sampling of 7°, and allowing shifts of 32 Å, and refinements were gradually improved by decreasing the binning factor, using less stringent low-pass filters, and finer angular sampling. Final refinement steps were carried out on unbinned data extracted in 224 voxel boxes, using a low-pass filter set at 7.4 Å, angular sampling of 1°, and shift limits of 7 Å. After 13 iterations convergence had been achieved and the refinements stopped. Low-pass filtering throughout all iterations was chosen based on FSC curves between independent half datasets, with a pixel-cut-off set at resolutions lower than indicated at the FSC 0.5 threshold. A box-wide soft saddle-shaped mask that excluded most of the membrane and the disordered outer coat was used during all alignment iterations. A total of 87,952 subtomograms (43,484 and 44,468 for each half) contributed to the final average. The two half-averages were weighted for the amplitude of the combined CTFs. Mask-corrected resolution assessment was carried out within the RELION[30] post-processing framework using a soft-edged mask around the central Sec23–Sec24–Sar1 trimer (Supplementary Figure 2), yielding a resolution of 4.87 Å at the 0.143 FSC cut-off. Local resolution estimation and local filtering were applied using RELION LocalRes, with a manually estimated B-factor of −350.

**Fitting**. X-ray structures of Sec23–24 and Sec23–Sar1 (1M2O and 1M2V) were fitted as rigid bodies into the density for each heterotrimer using UCSF Chimera "fit in map"[31]. Each protein subunit was then refined as individual rigid body. For Fig. 4c, d, PDB 2QTV and 2KYW were overlapped to the fitted model based on the common Sec23 subunit, using UCSF Chimera match-maker.

To calculate the difference map in Fig. 4, each subunit in the map was fitted with the corresponding X-ray model. Where only partial density was visible for subunits at the edges, the model was created based on the helical arrangement of the neighboring subunits. A map was generated from the fitted model using UCSF chimera, and low-pass filtered to 5 Å. The difference between normalized cryo-EM map and model-derived map was visualized in UCSF Chimera and Gaussian-filtered with a standard deviation of 3 pixels. A summary of the fitted model is given in Supplementary Table 2.

**Code availability**. Ad hoc scripts are available from the corresponding author upon request.

## Data availability

The cryo-EM map and fitted model have been deposited with accession number EMDB-0044 and PDB ID 6GNI. Other data are available from the corresponding author upon reasonable request.

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

## Acknowledgements

We thank Dr. Maya Topf and Dr. Carolyn Moores for discussion and Dr. Anthony Roberts and Dr. Alessandro Costa for comments on the manuscript. We thank Wim Hagen at the EMBL cryo-EM facility in Heidelberg for cryo-tomography data collection. We thank Dr. Charles Barlowe for the anti-Erv46 antibody, and the MRC LMB Mass Spectrometry facility for analysis of protein preparations. J.H. is supported by a Wellcome Trust PhD studentship (109161/Z/15/A), G.Z. is supported by a Royal Society Dorothy Hodgkin Fellowship (DH130048), V.S. and E.A.M. are supported by the Medical Research Council (MC_UP_1201/10). Data collection has been supported by iNEXT, project number 653706, funded by the Horizon 2020 programme of the European Union.

## Author contributions

J.H. performed GUV budding, made cryo-EM grids, and processed the data. V.S. and E.A.M. performed liposome binding, microsome budding, and yeast viability experiments. G.Z. designed the project, processed the data, and wrote the manuscript. All authors contributed to experimental design and to writing the manuscript.
