## [Peer Review File · Nature Communications]

Reviewers' comments:

Reviewer #1 (Remarks to the Author):

In this study, Hutchings et al. present the cryo-tomography (averaged) structure of membrane bound COPII. This is a fantastic experimental system, and the paper presents a nice jump in resolution from the previous Zanetti/Briggs work. However, the resolution is not quite sufficient for model refinement or for a detailed analysis of some of the new structural observations. The visualization of the Sar1 membrane anchor is a very nice advance.

I have 2 points of criticism, one regarding experimental details, the other on interpretation, that I feel the authors need to address. Some minor comments are also appended.

1 – I am a bit puzzled by the use of the his-tagged protein, Sec31-Nhis, and have several questions:

- As I look at the model of the COPII cage, I have to disagree with the author's statement "the position of the N-terminal His-6 should perturb stability of the beta-propeller interactions". The authors should clarify the basis for their structural point of view.

- The assertion is that Sec31-Nhis produces a less well-ordered outer cage. I accept this given the high expertise of the authors. However, the differences in Fig 1A and 1B, right panels, are subtle. I feel that some sort of quantitative/statistical assessment is required here.

- No reason was given explicitly as to why the Sec31-Nhis protein was used for the study. Does it allow for higher resolution than wild type protein? Clarification of this would be helpful for the manuscript.

- The Fig 1C and D experiments, regarding the yeast phenotype and in vitro budding reactions are surprising. The authors surmise a "kinetic assembly defect", but I'm sure they realize that the results are more difficult to explain than that. The his-tagged protein causes no growth phenotype, yet no vesicles form (really background level) in the budding reaction with GTP. These two results seem mutually incompatible. Forgive my suggestion, but I do recommend that the authors double check for any phosphatase contamination in the Sec31-Nhis protein prep. The wild-type Sec31 protein benefitted from an additional nickel-column step....

2 – The authors describe the Sar1 membrane anchor as inducing curvature. I think this is a very difficult issue that is not settled in the literature. Important work from Lee et al. & Schekman showed Sar1 at high density curving membrane. But, at low density (as in the COPII-associated array) its effects are likely to be modest; this statement based on the formative studies from Stephen White's lab on melittin, showing modest effects on lipid perturbation (e.g. *Biophys J*, 2001, Vol 80: 801-811). Further, Lee et al. were able to induce membrane curvature using a COPII complex in which the Sar1 anchor was removed (replaced with a 6-His tag). In my view, ascribing membrane curvature to any one of the COPII components is hardly possible. Hence I believe the authors should be circumspect in their interpretations as they cannot be certain whether their Sar1 membrane anchor is inducing curvature, or is responding to (and aligning along an axis of) curvature. This does not detract from the very nice visualization of the Sar1 anchor in the study.

Minor comments:

- Is "intercalate" the best way to describe what Sec31 is doing? I ask for two reasons. First, crystallographers and DNA biochemists often use intercalate to mean a substantial insertion into the lattice, whereas Sec31 seems more to be "organizing" the 23/24 lattice. Second, the interpretation that Sec31 "directly contributes to lateral interactions" really cannot be made at this resolution. To the eye, the PPP of Sec31 seems a little distant from the Sec23 neighbor – and this emphasizes the point that the statement of a "direct contribution" really needs to be backed up with some distance measurements, not quite possible at the current resolution.

- I respectfully suggest changing the statements in the intro: "hitherto elusive" and "challenging our current understanding of the mechanisms" – this is a bit magniloquent, and doesn't really fit with what is a very thoughtful and well-executed biophysical study. In fact, I feel that "hitherto elusive" gives insufficient emphasis to the new observation of the Sar1 membrane anchor. How about "We directly visualize, for the first time, the membrane-embedded Sar1 amphipathic.....".

Reviewer #2 (Remarks to the Author):

In this study Zanetti and colleagues apply cryotomography to determine structures of the inner-layer COPII coat (Sar1-Sec23-Sec24) when the outer layer cage (Sec13-Sec31) is fully assembled or when the Sec31 protein contains an N-terminal 6-His tag, which interferes with cage assembly but is still recruited to the inner-layer coat. Surprisingly, this altered version of Sec13-Sec31 imposes order on the inner layer and produced tightly coated membrane tubules without forming a full Sec13-Sec31 cage lattice. Generation of this presumptive COPII intermediate provided the investigators an opportunity to determine the structure (4.9 Å average resolution) and examine how Sec13-Sec31 promotes assembly of the inner coat lattice. Several important observations emerged from their analysis. First, Sar1-GTP molecules insert N-terminal amphipathic helix residues into the outer leaflet, take a sharp bend, and produce parallel wedge arrays that likely induce membrane tubules. The authors propose that regulating Sar1-GTP levels in COPII budding events controls the degree of membrane tube vs vesicle. Second, the cryo-EM map revealed regions where Sec31 density was continuous with inner coat proteins. Density was observed where Sec31 crosses the Sar1-Sec23 interface within a catalytic segment of Sec31 known to stimulate Sec23 GAP activity toward Sar1. Additional Sec31 density was observed between adjacent Sec23 molecules in the gelsolin-like domain that binds proline rich domains of Sec31 and other COPII regulators (e.g. TANGO1, cTAGE5).

Based on these findings, the authors propose that Sec13-Sec31 orders the inner layer coat through simultaneous contacts with Sec23 (catalytic region and gelsolin-like domain). A fully assembled Sec13-Sec31 cage is not required for this organization. In addition, regulation of Sar1-GTP hydrolysis by Sec31 and other factors (TANGO1, Sec16) are proposed to control membrane curvature of COPII structures.

Overall this is strong study that provides new insights based on determined cryo-EM structures. A couple of considerations are listed below.

1. The authors could directly test if Sec31 contact(s) with the Sec23-Sar1 interface and/or gelsolin-like domain is required for organizing the inner layer arrays. Key residues in Sec23 (including the disease linked F380) involved in binding the active segment of Sec31 have been mapped by structural studies. In addition, critical residues in the Sec23 gelsolin-like domain (Y672, Y678) have been mapped for binding to proline rich domains. Perhaps beyond the scope of this current study, but mutations in one or more of these regions would provide further insight on how Sec13-Sec31 organizes the inner layer.

2. The model for regulation of COPII carrier size (Fig. S7) is attractive although it may be an oversimplification. If locked Sar1-GTP drives membrane tubule formation one might not expect reconstituted COPII budding reactions from microsomes or liposomes (SUVs) supported with GMP-PNP to produce smaller sharply curved vesicles as has been documented. In other words, how are parallel arrays of Sar1-GTP incorporated into small vesicles?

Reviewer #3 (Remarks to the Author):

Hutchings et al., have used the technique of electron cryo-tomography and sub-tomogram averaging to investigate how the assembly of the COPII coat proteins influences membrane shape. The authors expressed and purified individual coat proteins and mixed them with liposomes to recreate the COPII coat. They discovered that: 1) the addition of a his-tag on sec31 prevented an ordered array being formed by the outer coat proteins, 2) an ordered array of outer coat proteins

in not required for an ordered array of inner coat proteins to be formed and 3) the inner coat proteins only form arrays if the outer coat proteins are present. To investigate how this might occur, they performed subtomogram averaging on the inner coat arrays which contained the his-tagged sec31 but no outer coat protein array. The resulting average had a resolution of 4.9 Å which by itself is a remarkable achievement. From inserting the known atomic models of the various coat proteins they have been able to predict the structure of the N-terminal amphipathic helix and assess how it induces membrane curvature, as well as predict the likely location of sec31 interaction with the inner coat proteins and how it promotes array formation.

The research presented by the authors has been well executed but I felt the text was confusing and I was not always sure what the authors had discovered and what was already known. It was also disappointing that the authors had only dedicated one sentence to quality of the subtomogram average that was obtained. As far as I know this is only the second subtomogram average which has reached a resolution beyond 6 Å and the first outside of the Brigg's lab. This is certainly the most exciting part of the paper and the authors should emphasize this more rather than just as a passing comment. Based on the quality of the subtomogram average alone, this manuscript should definitely be published in NCOMMS, but it would be more impactful if the text was clearer to the general reader.

Point-by-point comments:

Title: is very vague. It should at least say what type of coat protein was being studied and the resolution of the subtomogram average. E.g. "4.9 Å subtomogram average reveals how COPII proteins dictates membrane shape".

Abstract: line 9: "Here we present a 4.9 Å subtomogram average of the membrane-bound COPII inner coat proteins". It is probably worth also saying the authors are performing in vitro reconstitution of COPII proteins from yeast.

First paragraph could benefit from restructuring and clearer description of COPII proteins. E.g. telling the reader in the first sentence that COPII coated vesicles consist of two layers, an inner and an outer. Then describe what each layer consists of.

Second paragraph: I found this confusing as to what was already known and what the authors discovered. E.g. "Formation of Sar1-Sec23-Sec24 arrays depends on the outer coat" Is this a conclusion from the authors based on their results or is this already known from the literature. If the former the authors might want to rephrase the paragraph saying that they mixed different combinations of proteins together and from these results they were able to conclude the following.

Fig S2: I can't see the coat proteins on these tubes. A zoomed in view is required. Also an image of the liposomes without protein present needs to be presented as a negative control.

Why was the sec31-His sample used for the subtomogram averaging? This is not clear in the manuscript and needs to be explained.

Please elaborate on the subtomogram average. E.g. how does this compare to other structures generated by the Brigg's group. Were the same features described by the current authors present in other structures and if not, why not?

N-terminal amphipathic helix. After the authors describe the shape of the N-terminal amphipathic helix, they need to emphasize that all the Sar1 proteins have the same orientation in the array which means all the helices point in the same direction which is consistent with the direction of membrane curvature based on the knowledge that an amphipathic helix can generate membrane curvature. Also, as no classification was performed, the authors need to describe how confident

they are that the N-terminal helix only has one orientation in all the Sar1 proteins.

FIG 4a Please can the authors color code the difference densities and maybe provide an image 90° rotated to the membrane plane as it is not clear which density is which from the text or figure legend or the different heights of the densities.

Does ref8 define the interaction of sec31 to sec23 using x-ray crystallography or is this interaction a new discovery based on the current work. The text need to be rephrased to make it clearer.

A movie of the subtomogram average showing all the relevant parts would be appreciated by the reader.

Materials and Methods

Why does the name of proteins have a p in the methods but not in the text?

What is BR (page 14)?

In Sar1p, line 5, remove in from “was carried out in overnight while dialyzing”

Electron cryo-tomography data collection (iNEXT and Polara). iNEXT and Polara not required. If the authors keep this, they need to explain what iNEXT is.

Page 18: line 13 in paragraph on particle picking: remove the first “used” from “we used adjusted the CC values used for thresholding”

Page 18: line 21. Please add ‘s’ to subtomogram.

We would like to thank the reviewers for insightful comments on our manuscript. We have taken all reviewers' concerns into consideration and we are convinced that as a result our manuscript is significantly improved. Please find below a point-by-point response to the reviewers.

Reviewer #1 (Remarks to the Author):

In this study, Hutchings et al. present the cryo-tomography (averaged) structure of membrane bound COPII. This is a fantastic experimental system, and the paper presents a nice jump in resolution from the previous Zanetti/Briggs work. However, the resolution is not quite sufficient for model refinement or for a detailed analysis of some of the new structural observations. The visualization of the Sar1 membrane anchor is a very nice advance.

I have 2 points of criticism, one regarding experimental details, the other on interpretation, that I feel the authors need to address. Some minor comments are also appended.

1 – I am a bit puzzled by the use of the his-tagged protein, Sec31-Nhis, and have several questions:

- As I look at the model of the COPII cage, I have to disagree with the author's statement "the position of the N-terminal His-6 should perturb stability of the beta-propeller interactions". The authors should clarify the basis for their structural point of view.

We have clear evidence that the stability of the Sec31 vertex interaction is disrupted in Sec31-Nhis, but the reviewer is right that we cannot be sure that this is due to 'perturbation of the stability of the beta-propeller interactions', and not to destabilisation of the beta-propeller themselves.

We have now revised the relevant section in the paper; after discussing the evidence that N-his disrupts order in the outer coat, we propose these alternative explanations: *"On these tubes the outer coat was disordered: in contrast to the ordered lozenges seen with wild type Sec31, we could clearly visualize rods that appeared to "float", tethered to the budded tubule without forming regular arrays (Fig. 1B and S1D). We hypothesize that the peptide tag perturbs stability of the Sec31 N-terminal β -propellers or their interaction interface, and thus disfavors cage formation."*

- The assertion is that Sec31-Nhis produces a less well-ordered outer cage. I accept this given the high expertise of the authors. However, the differences in Fig 1A and 1B, right panels, are subtle. I feel that some sort of quantitative/statistical assessment is required here.

The negative stain overviews in Fig. S1 A and D show the difference between the tube's outer coat clearly, and the power spectra provide statistical assessment to the presence or absence of lattices. We have added distance measurement to confirm peaks in the power spectra correspond to outer and inner coat respectively in panels A and D of Fig. S1.

- No reason was given explicitly as to why the Sec31-Nhis protein was used for the study. Does it allow for higher resolution than wild type protein? Clarification of this would be helpful for the manuscript.

In the process of optimising budding reactions for cryo-EM grids, we noticed the Sec31-Nhis was better suited to pursue high-resolution structure determination than wild type Sec31. Visualisation of budding reactions using wild type Sec13-Sec31 showed that most of Sec13-Sec31 assembled in cages. This probably led to lower frequency of budding,

with fewer tubes visible on grids. Also, tubes were often inconveniently in contact with cages. Given the necessity for high-throughput data collection we have decided to optimise cryo-grids with the mutant N-his Sec31. We have now added an explanation as to why we used the Nhis protein to the materials and methods section.

- The Fig 1C and D experiments, regarding the yeast phenotype and in vitro budding reactions are surprising. The authors surmise a “kinetic assembly defect”, but I’m sure they realize that the results are more difficult to explain than that. The his-tagged protein causes no growth phenotype, yet no vesicles form (really background level) in the budding reaction with GTP. These two results seem mutually incompatible. Forgive my suggestion, but I do recommend that the authors double check for any phosphatase contamination in the Sec31-Nhis protein prep. The wild-type Sec31 protein benefitted from an additional nickel-column step.....

We have checked for presence of contaminant proteins by mass spectrometry, as the reviewer suggested, and did not see any significant phosphatase contamination, both in the wild type and Sec31-Nhis protein preparations. We agree that the results seem incompatible, and were also surprised. We speculate that viability of the N-his construct is supported by cellular factors that might stabilise coat assembly to a certain extent, sufficient for budding. We now discuss this point in the manuscript.

“Sec31-Nhis supported in vitro budding from microsomes in the presence of GMP-PNP, but in the presence of GTP failed to generate vesicles (Fig. 1D). Thus, under conditions that permit dynamic coat turnover, a stable outer coat assembly is required for productive budding (Fig. 1D). Interestingly, Sec31-Nhis supported viability in yeast (Fig. 1C), suggesting that additional factors may stabilize the coat in cells.”

2 – The authors describe the Sar1 membrane anchor as inducing curvature. I think this is a very difficult issue that is not settled in the literature. Important work from Lee et al. & Schekman showed Sar1 at high density curving membrane. But, at low density (as in the COPII-associated array) its effects are likely to be modest; this statement based on the formative studies from Stephen White’s lab on melittin, showing modest effects on lipid perturbation (e.g. Biophys J, 2001, Vol 80: 801-811). Further, Lee et al. were able to induce membrane curvature using a COPII complex in which the Sar1 anchor was removed (replaced with a 6-His tag). In my view, ascribing membrane curvature to any one of the COPII components is hardly possible. Hence I believe the authors should be circumspect in their interpretations as they cannot be certain whether their Sar1 membrane anchor is inducing curvature, or is responding to (and aligning along an axis of) curvature. This does not detract from the very nice visualization of the Sar1 anchor in the study.

This is an important point. We have now added this point to our discussion, and added supplementary figure panels (Suppl Fig5 C and D) to illustrate membrane interaction from the other COPII subunits.

Updated text below:

“This orientation is compatible with models proposing that amphipathic helices embed shallowly into lipid bilayers, creating a wedge to induce and/or sense membrane curvature¹⁷. Membrane curvature likely derives from Sar1 amphipathic helix insertion, as well as interaction with the membrane-proximal surface of Sec23-Sec24. Sec24 in particular appears to extensively contact the membrane through a largely basic surface that encompasses its Zn finger domain (Fig. S5C-D). Formation of ordered arrays of inner

coat thus leads to many Sar1 helices inserting parallel to each other (Fig. 3C) and to the inner surface of Sec23-Sec24 to adopt a cylindrical profile, reinforcing local membrane curvature and favoring tubes. This might in turn be sensed by additional Sar1 and Sec23-Sec24 subunits, which are efficiently recruited to the site of budding to further promote tubulation.”

Minor comments:

- Is “intercalate” the best way to describe what Sec31 is doing? I ask for two reasons. First, crystallographers and DNA biochemists often use intercalate to mean a substantial insertion into the lattice, whereas Sec31 seems more to be “organizing” the 23/24 lattice. Second, the interpretation that Sec31 “directly contributes to lateral interactions” really cannot be made at this resolution. To the eye, the PPP of Sec31 seems a little distant from the Sec23 neighbor – and this emphasizes the point that the statement of a “direct contribution” really needs to be backed up with some distance measurements, not quite possible at the current resolution.

We have changed ‘intercalate’ with ‘bridge’. We agree that the fitted atomic model shows the distance between PPP and neighbour Sec23 to be too great for direct interaction, nevertheless at the resolution achieved it is very clear that the density projecting from the PPP peptide and that of the neighbouring Sec23 are interconnected (see for example Fig S5A). We have changed the text to make it clearer that the contribution to lateral interactions is mediated by the PPP and surrounding residues, rather than strictly the PPP residues only. New text below:

“Our assembled structure reveals that the PPP motif and adjacent residues forms a bridge connecting neighboring Sec23 subunits: the PPP residues are part of a more extended region of density, which appears connected to both Sec23 molecules. The resolution in this area is lower than the rest of Sec23 (Fig. S3), likely because different PPP-containing regions (yeast Sec31 has seven such motifs) are bound to distinct Sec23 molecules.”

- I respectfully suggest changing the statements in the intro: “hitherto elusive” and “challenging our current understanding of the mechanisms” – this is a bit magniloquent, and doesn’t really fit with what is a very thoughtful and well-executed biophysical study. In fact, I feel that “hitherto elusive” gives insufficient emphasis to the new observation of the Sar1 membrane anchor. How about “We directly visualize, for the first time, the membrane-embedded Sar1 amphipathic.....”.

We have changed this as requested.

Reviewer #2 (Remarks to the Author):

In this study Zanetti and colleagues apply cryotomography to determine structures of the inner-layer COPII coat (Sar1-Sec23-Sec24) when the outer layer cage (Sec13-Sec31) is fully assembled or when the Sec31 protein contains an N-terminal 6-His tag, which interferes with cage assembly but is still recruited to the inner-layer coat. Surprisingly, this altered version of Sec13-Sec31 imposes order on the inner layer and produced tightly coated membrane tubules without forming a full Sec13-Sec31 cage lattice. Generation of this presumptive COPII intermediate provided the investigators an opportunity to determine the structure (4.9 Å average resolution) and examine how Sec13-Sec31 promotes assembly of the inner coat lattice. Several important observations emerged from their analysis. First, Sar1-GTP molecules insert N-terminal amphipathic helix residues into the outer leaflet, take a sharp bend, and produce parallel wedge arrays that likely induce membrane tubules. The authors

propose

that regulating Sar1-GTP levels in COPII budding events controls the degree of membrane tube vs vesicle. Second, the cryo-EM map revealed regions where Sec31 density was continuous with inner coat proteins. Density was observed where Sec31 crosses the Sar1-Sec23 interface within a catalytic segment of Sec31 known to stimulate Sec23 GAP activity toward Sar1. Additional Sec31 density was observed between adjacent Sec23 molecules in the gelsolin-like domain that binds proline rich domains of Sec31 and other COPII regulators (e.g. TANGO1, cTAGE5).

Based on these findings, the authors propose that Sec13-Sec31 orders the inner layer coat through simultaneous contacts with Sec23 (catalytic region and gelsolin-like domain). A fully assembled Sec13-Sec31 cage is not required for this organization. In addition, regulation of Sar1-GTP hydrolysis by Sec31 and other factors (TANGO1, Sec16) are proposed to control membrane curvature of COPII structures.

Overall this is strong study that provides new insights based on determined cryo-EM structures. A couple of considerations are listed below.

1. The authors could directly test if Sec31 contact(s) with the Sec23-Sar1 interface and/or gelsolin-like domain is required for organizing the inner layer arrays. Key residues in Sec23 (including the disease linked F380) involved in binding the active segment of Sec31 have been mapped by structural studies. In addition, critical residues in the Sec23 gelsolin-like domain (Y672, Y678) have been mapped for binding to proline rich domains. Perhaps beyond the scope of this current study, but mutations in one or more of these regions would provide further insight on how Sec13-Sec31 organizes the inner layer.

We agree that structure-based mutational studies of key residues involved in coat assembly would help clarify the role of the outer coat in organisation of the inner layer, and in general the role of coat interfaces in COPII budding. We have already embarked on mutational analysis of the sites suggested as well as other key contact interfaces. Our preliminary data suggests that mutations designed to disrupt individual interfaces have little effect on COPII budding, while their combination differently affects the coat behaviour in a number of assays. This is probably due to the fact that COPII assembly involves a rather intricate network of partially redundant interactions. We are working towards the hypothesis that this redundancy might serve to add plasticity to the COPII budding process and allow for fine regulation. Due to the complex picture emerging from our preliminary functional studies, we agree with the reviewer that the analysis of mutants extend beyond the scope of the current paper, and we reserve to report those results in a follow up study, when complete.

2. The model for regulation of COPII carrier size (Fig. S7) is attractive although it may be an oversimplification. If locked Sar1-GTP drives membrane tubule formation one might not expect reconstituted COPII budding reactions from microsomes or liposomes (SUVs) supported with GMP-PNP to produce smaller sharply curved vesicles as has been documented. In other words, how are parallel arrays of Sar1-GTP incorporated into small vesicles?

A fundamental difference between ours and previous studies is that we did not include any centrifugation steps prior to imaging budded membranes, as a result we see both tubes and vesicles. Centrifugation might lead to breaking tubulated membranes into

small vesicles due to sheer forces, and tubes might pellet together with donor membranes as they tend to remain attached to them.

Also, we do not propose in our model that vesicles contain extended parallel arrays of inner coat, but rather that these extended arrays do not have the time to form as GTP hydrolysis leads to their disassembly soon after assembly. This is conveyed in the model figure (Fig. S7 in the reviewed manuscript). We think it is very important to make this point clearly so we have moved the figure to the main text (Fig 5 in the revised manuscript version).

We have also clarified the description of our model in the discussion:

“When GTP hydrolysis is permitted, inner coat lateral interactions are transient, yielding Sar1 amphipathic helices that orient randomly and thereby drive membrane curvature in all directions, resulting in a spherical vesicle with no extended inner coat lattice. Under conditions of limited GTP hydrolysis, inner coat assembly would predominate over disassembly, and more extended lattices with parallel orientation of multiple Sar1 helices would lead to a uniform direction of membrane curvature and thus formation of tubules in a reaction independent of outer coat stable assembly.”

Reviewer #3 (Remarks to the Author):

Hutchings et al., have used the technique of electron cryo-tomography and sub-tomogram averaging to investigate how the assembly of the COPII coat proteins influences membrane shape. The authors expressed and purified individual coat proteins and mixed them with liposomes to recreate the COPII coat. They discovered that: 1) the addition of a his-tag on sec31 prevented an ordered array being formed by the outer coat proteins, 2) an ordered array of outer coat proteins is not required for an ordered array of inner coat proteins to be formed and 3) the inner coat proteins only form arrays if the outer coat proteins are present. To investigate how this might occur, they performed subtomogram averaging on the inner coat arrays which contained the his-tagged sec31 but no outer coat protein array. The resulting average had a resolution of 4.9 Å which by itself is a remarkable achievement. From inserting the known atomic models of the various coat proteins they have been able to predict the structure of the N-terminal amphipathic helix and assess how it induces membrane curvature, as well as predict the likely location of sec31 interaction with the inner coat proteins and how it promotes array formation.

The research presented by the authors has been well executed but I felt the text was confusing and I was not always sure what the authors had discovered and what was already known. It was also disappointing that the authors had only dedicated one sentence to quality of the sub-tomogram average that was obtained. As far as I know this is only the second subtomogram average which has reached a resolution beyond 6 Å and the first outside of the Brigg's lab. This is certainly the most exciting part of the paper and the authors should emphasize this more rather than just as a passing comment. Based on the quality of the subtomogram average alone, this manuscript should definitely be published in NCOMMS, but it would be more impactful if the text was clearer to the general reader.

We thank the reviewer for pointing this out, we have now underlined this achievement in our text:

“We obtained the structure of the assembled inner COPII coat to 4.9 Å resolution, representing a significant advance on the previously solved structure of the assembled COPII coat⁴, and the highest resolution subtomogram average of a membrane-bound coat

to date.”

Point-by-point comments:

Title: is very vague. It should at least say what type of coat protein was being studied and the resolution of the subtomogram average. E.g. “4.9 Å subtomogram average reveals how COPII proteins dictates membrane shape”.

We have changed the title to: “Sub-tomogram averaging of COPII assemblies reveals how coat organization dictates membrane shape”. We feel stating the resolution would be misleading: 4.9Å is an average resolution, and that number does not convey important information.

Abstract: line 9: “Here we present a 4.9 Å subtomogram average of the membrane-bound COPII inner coat proteins”. It is probably worth also saying the authors are performing in vitro reconstitution of COPII proteins from yeast.

We have added this.

First paragraph could benefit from restructuring and clearer description of COPII proteins. E.g. telling the reader in the first sentence that COPII coated vesicles consist of two layers, an inner and an outer. Then describe what each layer consists of.

We have reformatted the manuscript according to Nature Communications guidelines, expanding the introduction. While doing this, we have addressed the reviewer suggestion.

Second paragraph: I found this confusing as to what was already known and what the authors discovered. E.g. “Formation of Sar1-Sec23-Sec24 arrays depends on the outer coat” Is this a conclusion from the authors based on their results or is this already known from the literature. If the former the authors might want to rephrase the paragraph saying that they mixed different combinations of proteins together and from these results they were able to conclude the following.

We have now introduced a clear division between sections: “introduction”, “results”, and “discussion”, where it is clear the information mentioned by the reviewer is a novel finding of our work.

Fig S2: I can’t see the coat proteins on these tubes. A zoomed in view is required. Also an image of the liposomes without protein present needs to be presented as a negative control.

The panels in Fig S2 are medium magnification maps and it is not possible (nor expected) to see the coat. The figure aims to provide an overview of budding morphology. The coat is visible in the high magnification tomograms in Fig.1, which have been collected on the same grids. We have now clarified this in the figure legend. Budding reactions were performed from GUVs rather than standard liposome, which are over 10µm in diameter. They are therefore too big for cryo-EM. Negative stain EM only showed large black areas which we do not wish to include in the manuscript. We note that a reaction without Sec13-Sec31 was included in Figure S1, which is a good

control to show that straight coated tubes only form in the presence of the full set of COPII proteins.

Why was the sec31-His sample used for the subtomogram averaging? This is not clear in the manuscript and needs to be explained.

We have responded to the same concern raised by reviewer #1 (third question).

Please elaborate on the subtomogram average. E.g. how does this compare to other structures generated by the Brigg's group. Were the same features described by the current authors present in other structures and if not, why not?

The resolution of the current subtomogram average (4.9Å) is significantly better than the one which was reported before by Zanetti and Briggs (~25Å). While we confirm the overall arrangement of the inner coat lattice, with this work we gain novel biological insight, in particular related to molecular interactions between lattice subunits, interaction between inner and outer coat, and amphipathic helix insertion.

We now address this comparison in the main text:

“We obtained the structure of the assembled inner COPII coat to 4.9 Å resolution, representing a significant advance on the previously solved structure of the assembled COPII coat¹, and the highest resolution subtomogram average of a membrane-bound coat to date. Our structure reveals in molecular detail how outer coat proteins promote inner coat assembly through their triple-proline motifs, and how parallel orientation of inner coat subunits and insertion of the Sar1 amphipathic helix promotes tubular curvature of budded membranes.”

N-terminal amphipathic helix. After the authors describe the shape of the N-terminal amphipathic helix, they need to emphasize that all the Sar1 proteins have the same orientation in the array which means all the helices point in the same direction which is consistent with the direction of membrane curvature based on the knowledge that an amphipathic helix can generate membrane curvature. Also, as no classification was performed, the authors need to describe how confident they are that the N-terminal helix only has one orientation in all the Sar1 proteins.

We feel this aspect was already emphasised in our results as well as in the discussion:

1. “Its clearly defined elongated shape indicates that Sar1 amphipathic helices all adopt roughly the same orientation with respect to the protein core, running along the long axis of the tube (Fig. 3C).”

2. “Formation of ordered arrays of inner coat thus leads to many Sar1 helices inserting parallel to each other (Fig. 3C)”

3. “Under conditions of limited GTP hydrolysis, inner coat assembly would predominate over disassembly, and more extended lattices with parallel orientation of multiple Sar1 helices would lead to a uniform direction of membrane curvature and thus formation of tubules.”

We are indeed looking at an average of particles, the fact that we see any directionality for the N-terminal Sar1 amphipathic helix means that helices are all roughly parallel. It is possible that the lower resolution at this site of the map is due to slight variations in helix direction. We attempted to address this by classification, unfortunately the number of particles is not sufficient to be able to sub-classify without losing resolution,

but we are confident that the helices are at angles similar enough to each other to produce a clearly distinguishable average.

FIG 4a Please can the authors color code the difference densities and maybe provide an image 90° rotated to the membrane plane as it is not clear which density is which from the text or figure legend or the different heights of the densities.

We have added a panel with a 90° view for the difference density, and have edited the figure to hopefully make it more clear: instead of colouring the densities (which would require segmentation of the difference map), we have boxed them in with different colours in the panels.

Does ref8 define the interaction of sec31 to sec23 using x-ray crystallography or is this interaction a new discovery based on the current work. The text need to be rephrased to make it clearer.

Ref 8 (now 9) defines the interaction between Sec23 and PPP peptides (obtained from TANGO1). Our contribution was to confirm structurally that Sec31 PPP bind at the same site, and that this binding acts as a bridge between lattice subunits. We have now clarified this as part of the more extended introduction, as well as in the section of the results dedicated to describing this interaction.

A movie of the subtomogram average showing all the relevant parts would be appreciated by the reader.

We have now included a movie to summarise our structure.

Materials and Methods

Why does the name of proteins have a p in the methods but not in the text?

We have now used the same nomenclature throughout.

What is BR (page 14)?

“Budding Reaction”, now spelled out.

In Sar1p, line 5, remove in from “was carried out in overnight while dialyzing”

Done

Electron cryo-tomography data collection (iNEXT and Polara). iNEXT and Polara not required. If the authors keep this, they need to explain what iNEXT is.

We have removed reference to iNEXT and polara.

Page 18: line 13 in paragraph on particle picking: remove the first “used” from “we used adjusted the CC values used for thresholding”

Done

Page 18: line 21. Please add 's' to subtomogram.

Done

REVIEWERS' COMMENTS:

Reviewer #1 (Remarks to the Author):

The authors have done a commendable job responding to the criticisms. I recommend publication in your Journal.

One remaining minor comment:

The observation that the EM density is seen to merge between the PPP motif and adjacent Sec23 is not evidence for contact; merger may just be a feature of the limiting resolution. (Indeed, merger of densities belonging to two independent objects is one way to measure resolution limit in optical imaging). Only distance measurements from a refinement could provide this structural information. Hence, I feel the authors should say that the merged density "suggests contact" rather than providing evidence.

Reviewer #2 (Remarks to the Author):

This revised manuscript is strengthened and satisfactorily addresses my initial comments. In my opinion this work is entirely appropriate for publication in Nature Communications.

Reviewer #3 (Remarks to the Author):

The authors have responded to all my comment adequately.

We would like to thank the reviewers for their comments on the revised manuscript entitled “Sub-tomogram averaging of COPII assemblies reveals how coat organization dictates membrane shape”. Please find below responses to the remaining comments.

REVIEWERS' COMMENTS:

Reviewer #1 (Remarks to the Author):

The authors have done a commendable job responding to the criticisms. I recommend publication in your Journal.

One remaining minor comment:

The observation that the EM density is seen to merge between the PPP motif and adjacent Sec23 is not evidence for contact; merger may just be a feature of the limiting resolution. (Indeed, merger of densities belonging to two independent objects is one way to measure resolution limit in optical imaging). Only distance measurements from a refinement could provide this structural information. Hence, I feel the authors should say that the merged density "suggests contact" rather than providing evidence.

As requested, we have changed the sentence:

“Thus, our structure provides direct evidence that Sec31 binds to Sec23 through PPP motifs at the same site as TANGO1...” to “Thus, our structure suggests that Sec31 binds to Sec23 through PPP motifs at the same site as TANGO1...”

Reviewer #2 (Remarks to the Author):

This revised manuscript is strengthened and satisfactorily addresses my initial comments. In my opinion this work is entirely appropriate for publication in Nature Communications.

Reviewer #3 (Remarks to the Author):

The authors have responded to all my comment adequately.

FURTHER CHANGES:

In addition to the reviewer comments, we have also added “The membrane in panel D was segmented from a Gaussian filtered version of the structure.” to the legend for Figure 2. The full version of the legend for panels C and D is shown below:

“Surface representation of the locally filtered cryo-EM map (transparent white), with the atomic models (PDB 1M2O and 1M2V) fitted and colored according to the scheme in panel A. Top and side views, respectively. The membrane in panel D was segmented from a Gaussian filtered version of the structure.”